# Opioid-Induced Sexual Dysfunction in Cancer Patients

**DOI:** 10.3390/cancers14164046

**Published:** 2022-08-22

**Authors:** Bartłomiej Salata, Agnieszka Kluczna, Tomasz Dzierżanowski

**Affiliations:** Laboratory of Palliative Medicine, Department of Social Medicine and Public Health, Medical University of Warsaw, ul. Oczki 3, 02-007 Warsaw, Poland

**Keywords:** cancer, pain management, opioid, sexual dysfunction, sexual disorder, erectile dysfunction

## Abstract

**Simple Summary:**

Sexual disorders affect up to 80% of cancer patients, depending on the type of cancer, yet they are commonly overlooked and untreated. Opioid-induced sexual dysfunction (OISD) is reported in half of opioid users. The pathophysiology of OISD—still a subject for research—may include disorders of both the endocrine and nervous systems, expressed in, among other things, erectile dysfunction and declined sexual desire, sexual arousal, orgasm, and general satisfaction with one’s sex life. The etiology of sexual dysfunction in cancer patients is usually multifactorial, so the management should be multifaceted and individualized by targeting pathophysiological factors. The treatment options for OISD are few and include testosterone replacement therapy, bupropion, opioid antagonists, phosphodiesterase type 5 inhibitors, plant-derived substances, and non-pharmacological treatments, although the evidence is insufficient. One of the treatment options may also be a choice of an opioid that is less likely to cause sexual dysfunction, yet further research is necessary.

**Abstract:**

Sexual dysfunction is common in patients with advanced cancer, although it is frequently belittled, and thus consistently underdiagnosed and untreated. Opioid analgesics remain fundamental and are widely used in cancer pain treatment. However, they affect sexual functions primarily due to their action on the hypothalamus–pituitary–gonadal axis. Other mechanisms such as the impact on the central and peripheral nervous systems are also possible. The opioid-induced sexual dysfunction includes erectile dysfunction, lack of desire and arousal, orgasmic disorder, and lowered overall sexual satisfaction. Around half of the individuals taking opioids chronically may be affected by sexual dysfunction. The relative risk of sexual dysfunction in patients on chronic opioid therapy and opioid addicts increased two-fold in a large meta-analysis. Opioids differ in their potential to induce sexual dysfunctions. Partial agonists and short-acting opioids may likely cause sexual dysfunction to a lesser extent. Few pharmaceutical therapies proved effective: testosterone replacement therapy, PDE5 inhibitors, bupropion, trazodone, opioid antagonists, and plant-derived medicines such as Rosa damascena and ginseng. Non-pharmacological options, such as psychosexual or physical therapies, should also be considered. However, the evidence is scarce and projected primarily from non-cancer populations, including opioid addicts. Further research is necessary to explore the problem of sexuality in cancer patients and the role of opioids in inducing sexual dysfunction.

## 1. Introduction

Sexuality is an essential aspect of life, also for cancer patients [1]. Despite this, many of them believe that they do not receive proper care in this sphere of life [2], and only one in ten cancer patients are asked by a doctor about the quality of their sex life [3]. The quality of the sexual life of these patients has often deteriorated, and there are various reasons for this, such as pain and other physical symptoms, deformities of the body due to cancer or after medical interventions, the feeling of being unattractive, medications, or the lack of privacy conditions in long-term care facilities [4,5,6]. As opioids are often used in this group of patients, their influence on libido and the quality of sexual function is essential. This publication aims to synthesize the current knowledge on the relationship between the use of opioids and the occurrence of sexual dysfunctions.

## 2. Etiology and Pathophysiology

The etiology of sexual dysfunctions is often multifactorial, making it challenging to distinguish these dysfunctions and then clearly set a proper diagnosis. The fundamental problem is whether a given disorder is due to organic dysfunction or is psychological in origin. The International Classification of Diseases 11th Revision (ICD-11) divides sexual disorders into sexual dysfunctions and sexual pain disorders [7]. Additional coding (HA40.2) is recommended for the conditions associated with the use of opioids (Table 1).

In cancer patients, occurrence of sexual dysfunctions depends on the primary tumor location and treatment used. It is associated with, among other things, damage to the vascularization or innervation of the genital organs and their damage or post-surgical scarring, radiotherapy, high-dose chemotherapy, hormonal disorders, chronic fatigue and pain [6,8]. It results in (1) loss of desire; (2) genitourinary atrophy, dryness and pain, (3) difficulty experiencing pleasure and reaching orgasm, and (4) erectile dysfunction. Additionally, they may also be a sequela of psychological problems, such as unacceptance of one’s body image. On top of that, stomas in patients with gastrointestinal or urinary tract cancers may impede sexual activity as well.

### 2.1. The Role of Hormones

Hormones are an essential factor in modulating sexual functions. The role of testosterone, dehydroepiandrosterone (DHEA), and prolactin are best known, while the functions of estrogen, oxytocin, and progesterone are less clear. Testosterone in men increases libido, the degree of excitement, sexual satisfaction, the degree of penile stiffness, and the time of erectile response [9]. In women, it increases desire, excitement, vaginal congestion, and orgasm. These effects in women may, to some extent, be the effect of testosterone conversion to dihydrotestosterone and estradiol [10,11].

DHEA, produced by the adrenal glands, works mainly as a prohormone, and testosterone, dihydrotestosterone, estrone, and estriol are formed as a result of its subsequent transformations. It has a positive effect on desire, arousal, frequency of sexual thoughts, and satisfaction with the physical and emotional aspects of sexuality, among other things [12].

Prolactin most likely inhibits sexual functions. It delays ejaculation and reduces craving. Its concentration increases after the onset of orgasm, and it is responsible for the subsequent refractory period (feeling of sexual satiety and inhibited sexual behavior) [13]. Estrogens affect proper vaginal lubrication in women, increase excitement and, through muscle relaxation, prevent dyspareunia [14]. Oxytocin at high doses probably reduces sexual arousal and causes a refractory period after orgasm in males, whereas, at lower doses, it probably stimulates sexual behavior. Its secretion increases during sexual arousal and probably also stimulates the occurrence of erection—its concentration in the central nervous system is reduced in men with erectile dysfunction [15]. Progesterone is poorly understood in this respect. It probably inhibits sexual functions [14].

The effects of opioids on the endocrine system are probably mainly related to their effects on the hypothalamic–pituitary–gonadal axis (Figure 1). The mu-opioid receptors (MOR) are found in the hypothalamus [16], pituitary [17], testes [18], and ovaries [19]. Therefore, inhibition by opioid agonists of both the hypothalamus’s pulsatile secretion of the gonadotropin-releasing hormone (gonadoliberin; GnRH) causing hypogonadotropic hypogonadism, and the testosterone secretion directly in the testes occur. Another mechanism is related to the increase in prolactin secretion by the pituitary gland, which reduces the secretion of testosterone [20]. In addition, the production of DHEA in the adrenal cortex may be reduced [20], and a study on rats in which morphine was used shows that there may be an increase in the mRNA expression of enzymes that break down testosterone [21].

### 2.2. Sexual Desire

Sexual desire can be defined as a subjective psychological state related to the initiation and maintenance of sexual behavior caused by internal or external factors [22] or as the sum of factors that motivate or demotivate a person to engage in sexual activity [23]. It depends on biological, psychological, and social aspects. The biological aspects comprise, inter alia, the actions of the endocrine system and neurotransmitters. The stimulating neurotransmitters include dopamine, norepinephrine, oxytocin and melanocortins (beta-endorphin, adrenocorticotropic hormone, and alpha-melanotropin). The inhibitory ones include serotonin, endogenous cannabinoids, and opioids [24]. In the DSM-5 (Diagnostic and Statistical Manual of Mental Disorders) classification, there is a concept of Female Sexual Interest–Arousal Disorder (FSIAD), defined as the absence or a significant reduction in sexual interest or arousal. It consists of six domains: (1) sexual activity, (2) sexual or erotic thoughts or fantasies, (3) initiation of sexual activity, (4) excitement or pleasure during sexual activity, (5) sexual interest or arousal, and (6) genital or non-genital sensations during sexual activity. The absence, or a significant reduction, of at least three of them for at least six months and 75–100% of the time allows the diagnosis of the disorder [25].

In men, a similar disorder, in the DSM-5 classification, is the reduction of sexual desire in men, defined as Male Hypoactive Sexual Desire Disorder (MHSDD) [25].

### 2.3. Erection

Male erection is a complex neurovascular process dependent on balancing inhibitory and stimulating factors (sympathetic and parasympathetic systems, respectively). The reflex is caused by the stimulation from the sacral (S2–S4) section of the spinal cord, triggered by stimulation of penile afferents or by higher centers of the central nervous system upon visual, olfactory, and tactile stimulation, or imagination [26,27,28]. It causes the extension of the penile arteries, and simultaneous pressure on the venous vessels causes blood stagnation in the corpus cavernosum of the penis and, as a result, an erection.

Erectile dysfunction (ED) in a man can be diagnosed when during almost all occasions of sexual activity (75–100% on average), at least one of the following three symptoms occurs: (1) marked difficulty in obtaining an erection during sexual activity, (2) marked difficulty in maintaining an erection until completion of sexual activity, (3) marked decrease in erectile rigidity [25]. The incidence of ED in the general population ranges from 10–15% in men aged 40–49 to 50–70% in men aged 60–79 years [29,30] Risk factors include age, cardiovascular diseases, diabetes, obesity, smoking, and depression [31,32]. The incidence of ED in the cancer patient population may be around 29% at the time of diagnosis and 43% after treatment, but it is strongly dependent on the type of cancer and may be up to 80–90% for prostate, anus or colorectal cancers [33]. The effect of opioids on male erection is not well known. Several mechanisms that may coincide are suggested. First of all, by affecting MOR located in the area of the paraventricular nucleus of the hypothalamus, these drugs can inhibit the synthesis of nitric oxide (NO), which is part of the neurological pathway responsible for inducing an erection [27,34]. Another mechanism is related to the aforementioned reduction of testosterone concentration by inhibiting GnRH secretion in the hypothalamus and directly inhibiting testosterone secretion in the testes [20]. Based on studies in animal models, it can be assumed that there are other mechanisms affecting the peripheral nervous system [35].

### 2.4. Orgasm

A woman’s orgasm can be defined as “a variable, transient peak sensation of intense pleasure creating an altered state of consciousness usually with an initiation accompanied by involuntary, rhythmic contractions of the pelvic striated circumvaginal musculature often with concomitant uterine and anal contractions and myotonia that resolves the sexually-induced vasocongestion (sometimes only partially) and myotonia usually with an induction of well-being and contentment” [36]. In men, orgasm is related to ejaculation and results from stimulation of the vulvar nerve by increasing pressure in the posterior part of the urethra during ejaculation, stimulation of verumontanum, and contraction of the urethra and accessory sexual organs [37]. Physiological changes during a male orgasm are similar to those in the female body: contractions of the pelvic muscles and anal sphincter, hyperventilation, tachycardia, and increased blood pressure [38]. An orgasmic disorder listed in the ICD-10 classification is inhibition of orgasm (anorgasmia). It occurs when orgasm is not achieved despite high levels of excitement, or the intensity of the feeling of orgasm is reduced or delayed. It is more common in women than men [25].

The data on a prevalence of orgasmic disorders in cancer patients and mechanisms of influence of opioids on orgasm is scarce. In women it may be connected with low testosterone level [20].

## 3. Diagnostic Tools

Apart from the regular history, the most frequently used tools for diagnosing and assessing the effectiveness of sexual disorders treatment are self-report techniques consisting of the patient’s self-assessment by answering the questions posed in a given questionnaire. They facilitate a conversation about the sexual domain of the patient’s life and consecutive diagnostics. They can be especially beneficial for practitioners inexperienced in collecting sexological history. None of the available tools refer to opioid use, nor have they been validated in the opioid users’ population. 

### 3.1. Female Sexual Function Index (FSFI)

For women, the Female Sexual Function Index (FSFI) can be used (Table 2). It consists of 19 items combined in six areas: (1) desire, (2) arousal, (3) lubrication, (4) orgasm, (5) sexual satisfaction, and (6) pain, assessed for the last four weeks [39]. This questionnaire has been validated in cancer survivors [40] and is suggested by the Cancer Patient-Reported Outcomes Measurement Information System (PROMIS) Sexual Function Committee [41]. Additionally, the Brief Sexual Symptom Checklist for Women (SSFF-A) can be used as a primary screening tool of women with cancer [42].

### 3.2. International Index of Erectile Function (IIEF)

In men, the International Index of Erectile Function (IIEF), which is also suggested by the PROMIS Sexual Function Committee [41], can be used. It is a 15-item questionnaire assessing five areas (Table 2) in the four weeks prior to testing [43]. The IIEF-5 questionnaire, i.e., a shortened version of the IIEF consisting of five items, may be more convenient in clinical practice [44].

Additionally, there are questionnaires developed for specific cancer populations: University of California-Los Angeles Prostate Cancer Index (UCLA PCI) [45], Sexual Function-Vaginal Changes Qeustionnaire [46] for the assessment after gynecological cancer and FSFI adaption for breast cancer patients (FSFI-BC) [47].

## 4. Epidemiology

### 4.1. Sexual Disorders in Cancer Patients

Sexual disorders are a common problem in cancer patients, and their occurrence depends on the primary tumor location and treatment used [6,8]. On average, they affect more than half of patients, but they vary widely depending on specific cancers. In female breast cancer, it may be around 66% [48], 65–90% in colorectal cancers [33,48], 78% in gynecological cancers [48] and up to 80% in prostate cancer [49].

### 4.2. Sexual Dysfunction in Patients Taking Opioids

There has been little research on sexual dysfunction with opioids in people with cancer as yet, so most of the evidence comes from studies in other patient populations.

In a case–control study by Rajagopal et al. [50], the incidence of hypogonadism and sexual dysfunction in men on chronic opioid therapy for cancer pain was assessed. The study and control groups consisted of 20 men each, who had taken at least 200 mg/day of an oral morphine equivalent (OME) dose for at least one year, or placebo, respectively. The total testosterone, FSH, and LH concentrations were measured, and the quality of sexual function was assessed using the Sexual Desire Inventory (SDI) questionnaire. The mean concentration of all three hormones was two to three times lower in the study than in a control group. The mean SDI score was 18.5 vs. 40 in the control group, and it was statistically significant (*p* = 0.01).

In a study by Venkatesh et al. [51], the sexual function of 100 men with a history of at least one year of opioid dependence vs. 50 men in the control group was assessed. Forty-eight percent, vs. eight percent in the control group, had sexual dysfunction according to the Arizona Sexual Experiences Scale (ASEX). Of them, 45% had ED, defined as less than 25 points on the IIEF-5 scale, significantly more often than in the control group (16%). Ninety-two percent of the study group had impairment of at least one of the five functions tested on the IIEF-5 scale, vs. sixteen percent in the control group. Other sexual functions were also impaired vs. the control group: desire (41% vs. 8%), sexual arousal (29% vs. 2%), the ability to achieve orgasm (21% vs. 0%), and satisfaction with orgasm (25% vs. 6%).

In a meta-analysis by Zhao et al. [52] of nine cross-sectional studies and one cohort study involving 8829 patients on chronic opioid therapy, or heroin- or opium-addicted, the relative risk (RR) of ED approached 2. In addition, a strong association between long-term opioid use (>3 years) and ED was reported (RR 2.25), also in men under 50 years (RR 2.21).

Deyo et al. [53] investigated the frequency of prescribing medications or testosterone replacement therapy (TRT) for ED in 11,327 men prescribed opioids for lower back pain. It appeared to be associated with the doses and duration of opioid use, and in the case of long-term opioid use (>120 days or at least ten prescriptions over >90 days), it equaled 13.1% and was higher (19%) in the presence of a high daily opioid dose (OME > 120 mg).

In a prospective observational study (Ajo et al. [54]), opioid-induced sexual disorders were reported in 33% of patients. ED was present in 27.6%, and in 64% of cases it was assessed as severe.

In a study on men by Rubinstein et al. [55], use of long-acting opioids was connected with higher frequency of hypogonadism than in men using short-acting opioids—74% (34/46) vs. 34% (12/35). After controlling for daily dosage and body mass index, men on long-acting opioids had 4.78 times greater odds of becoming hypogonadal than men on short-acting opioids. The studies on long-acting (sustained release) opioids seem to be of great importance, as opioids of that type are a base of cancer pain treatment.

#### 4.2.1. Tramadol

In a case–control study, Hashim et al. [56] compared sexual function in a group of opioid addicts in tramadol, heroin, and control groups of 30 patients each. The mean scores on the IIEF-5 scale regarding erection were 8.6, 15, and 29.9, respectively, and the differences between the groups were statistically significant (*p* < 0.001). Compared to placebo, tramadol worsened orgasm (*p* = 0.003), desire (*p* = 0.002), and overall satisfaction (*p* < 0.001). The concentrations of free testosterone (*p* = 0.041) and LH (*p* = 0.004) were also significantly reduced versus the control group. However, all the assessed indicators were significantly better among tramadol addicts than heroin users (*p* < 0.001).

In a small study by Kabbash et al. [57], ED occurred in 44% of addicts taking tramadol and 10% in the placebo control group (*p* = 0.001). The occurrence of ED was dose-related and equaled 14.3% in the individuals taking ≤ 400 mg/day, 48.4% in the case of 400–1000 mg/day, and 50% when the dose exceeded 1000 mg/day, although the differences between the groups were not statistically significant (*p* = 0.23). Notably, a higher incidence of ED was reported when the daily dose exceeded the maximum recommended for regular medical use. Furthermore, the incidence of ED depended on the duration of tramadol use and was 20% for 1–2 years, 30.4% for 2–5 years, and 63.6% for more than 5 years (*p* = 0.04). Serum testosterone concentration was significantly lower in tramadol addicts than in the control group (*p* = 0.001), whereas serum prolactin concentration was significantly higher (*p* = 0.001). Consistently, a higher incidence of decreased libido was noticed in the group taking tramadol than in the control group (48% vs. 16.7%; *p* = 0.005). Noteworthily, 20% of the individuals in this study reported off-label tramadol use for the prevention of premature ejaculation [57]. Based on animal models, low tramadol doses may stimulate ejaculation, while high doses have an inhibitory effect [58]. According to a few studies, it may be effective in treating premature ejaculation [59]; however, the quality of evidence is low. Interestingly, one of reported adverse events was erectile dysfunction.

#### 4.2.2. Morphine

So far, there is no clinical evidence of morphine concerning sexual dysfunction. However, based on animal models, the intraperitoneal administration of morphine to male rats reduces the likelihood of erections proportionally to its dose, and the effect was reversed by naloxone [60]. Interestingly, the administration of naloxone alone at the lowest dose tested (0.1 mg/kg) also inhibited erection, and this effect was not observed at higher doses (1 and 10 mg/kg).

Additionally, the impact of morphine on sexual functions can be extrapolated from diamorphine (diacetylmorphine, heroine), a pro-drug deacetylated to morphine as an active molecule.

#### 4.2.3. Methadone and Buprenorphine

Methadone and buprenorphine, especially as an opioid substitution therapy, are collectively the subject of the largest number of research papers on this topic. In a meta-analysis of 16 studies on the prevalence of sexual dysfunction among male patients on methadone and buprenorphine therapy by Yee et al. [61], 52% of the 1570 methadone-treated patients reported sexual dysfunction. ED was assessed in 12 studies and occurred in 46% of patients. Decreased desire or libido was observed in 51% of patients in four studies collectively.

In the meta-analysis by Zhao et al. [52], the use of methadone was associated with a lower risk of ED (RR = 1.82) compared to other opioids (heroin and opium; RR = 2.04) in this study, which may explain the improvement in sexual function after starting methadone replacement therapy.

In the above-cited meta-analysis by Yee et al. [61], the incidence of sexual dysfunction in patients taking buprenorphine was 24%. A meta-analysis of four studies comparing the incidence of these disorders in patients taking buprenorphine and methadone revealed that methadone was associated with a five-fold higher risk of sexual dysfunction (OR = 4.95). Another study by this author [62] showed that patients taking buprenorphine (average dose 2.4 mg/day) reported a higher degree of sexual desire than patients using methadone (average dose 74.5 mg/day)—7.6 and 6.1 on the IIEF scale, respectively—and higher testosterone concentrations (18.5 vs. 12.5 nmol/L).

In turn, in a small study by Hallinan et al. [63], the mean IIEF score did not differ significantly between buprenorphine and control groups and was better than in the methadone-treated patients (61 vs. 50).

In a study of 258 women, mean age of 38 years, taking methadone (mean 61 mg/day) or buprenorphine (mean 11 mg/day) as opioid maintenance therapy, 56% of patients reported sexual dysfunction [64]. Notably, the patients with sexual dysfunction were characterized, inter alia, by older age, lower levels of education, higher doses of methadone, and worse mental health than patients without sexual dysfunction.

The lower incidence of sexual dysfunction in buprenorphine use compared to other opioids may be explained by its partial agonist/antagonist mode of action [65].

#### 4.2.4. Tapentadol

In a 12-week randomized clinical study, Baron et al. [66] compared the safety of prolonged-release tapentadol with prolonged-release oxycodone/naloxone tablets in patients with severe chronic low back pain with a neuropathic component. The participants were <64 years old and had an initial testosterone concentration within the normal range. Mean doses of opioids were 362 mg for tapentadol and 83 mg for oxycodone/naloxone. In 45.5% of the oxycodone/naloxone patients, testosterone concentrations decreased below the norm, while this only occurred in 10.5% of the tapentadol group.

An important limiting factor for the presented studies is a lack of control for confounding factors (e.g., pain, mental health, physical health, quality of life) between opioid and non-opioid groups. Additionally, most of the evidence is derived from patients addicted to opioids, and this population may not be representative for patients taking opioids for cancer pain as addiction differs from appropriate prescription opioid use and both populations are also different.

## 5. Treatment of Opioid-Induced Sexual Dysfunction

There are few treatment options for sexual dysfunction (Table 3), and most of the studies were based on male groups only. The choice of an opioid with a less-negative impact on the endocrine system, such as buprenorphine or tapentadol, seems to be a vital element of management, but clinical evidence for the effectiveness of such a strategy is scarce.

### 5.1. Testosterone Replacement Ttherapy

In a medium-sized, 3-year prospective observational study, Ajo et al. [54] verified the effectiveness of testosterone replacement therapy (TRT) and the phosphodiesterase type 5 inhibitor (PDE5i) for the treatment of ED in patients using opioids (the mean duration of opioid therapy was 5 years and 6 months, the mean opioid dose was 107.1 mg/day OME). After six months of therapy, 42% of patients experienced a significant improvement as measured by the IIEF questionnaire. A positive correlation was also observed between the improvement in the IIEF score and the quality of sexual life and a reduction of anxiety. However, one systematic review [67] suggests that TRT may be effective only in improving pain and emotional functioning, but not sexual function. The quality of evidence is low, and further research is needed.

### 5.2. Bupropion and Trazodone

A small-sized, randomized, double-blind, placebo-controlled trial investigated the efficacy of 50 mg bupropion twice a day in the treatment of sexual dysfunction in men using methadone (mean 70 mg for 46 months) [68]. The mean erection quality score measured on the IIEF-15 scale improved from 18.1 to 22.6, and sexual satisfaction from 7 to 8.8, with statistically significant differences, compared to the control group (*p* = 0.03 and *p* = 0.02, respectively).

The efficacy of trazodone on erectile dysfunction in men on methadone maintenance therapy was evaluated in a small study [69]. Patients received 50 mg/day of trazodone for four days, then the dosage was increased to 100 mg/day and maintained for six weeks. The mean score on the Erectile Dysfunction Intensity Scale (EDIS) improved from 12.21 to 16.78 (*p* < 0.05; 5–10—severe ED, 11–15—moderate ED, 16–20—mild ED, 21–25—no ED)

### 5.3. Opioid Receptor Antagonists

Another therapeutic option for ED is the use of an opioid receptor antagonist. In a study by van Ahlen et al. [70], the efficacy of naltrexone in the treatment of idiopathic ED has been investigated. Patients in the study group took 25 mg of naltrexone for four weeks followed by 50 mg of naltrexone for another four weeks. An improvement in the number of morning spontaneous erections was reported for both the 25 mg and 50 mg naltrexone treatment arms. Such an improvement was not seen in the placebo group. Neither libido, nor FSH, LH, or testosterone, changed in either the research or the placebo groups. The authors attribute the erectile-stimulating effect of naltrexone in the group of patients not taking opioids to the antagonization of endogenous opioids, which may inhibit sexual function and lower LH release. An alternative to naltrexone may be nalmefene, which may also increase FSH, LH, and testosterone levels [71].

In one study the frequency of sexual dysfunctions in patients on buprenorphine and naltrexone maintenance therapy was compared [72]. Erection difficulty and reduction of sexual desire was reported more often by patients in the naltrexone group than in the buprenorphine group (66.7% vs. 43.3% and 46.7% vs. 33.3%) when asked about experiencing them “ever”. However, when asked about the last month, both groups reported similar frequency of erection difficulty, and the reduction of sexual desire was higher in the buprenorphine group (10% vs. 26.7%). Additionally, sexual functions were similar between the two groups when measured with Brief Male Sexual Functioning Inventory (BMSFI) and asked about the last month. The quality of evidence is rather poor.

There are no studies on peripheral restricted opioid receptor antagonists as a treatment for opioid-induced sexual dysfunction.

### 5.4. Plant-Derived Medicines

There is limited evidence for the effectiveness of herbal products, including damask rose oil. It has been tested in a small, randomized, double-blind, placebo-controlled trial of women undergoing methadone replacement therapy [73]. Improvement was demonstrated in all domains of the FSFI scale after eight weeks of treatment.

Another randomized controlled trial shows that ginseng may also have a positive effect on methadone-induced sexual function, both in men and women [74].

### 5.5. Non-Pharmacological Methods

There are no studies on the effectiveness of non-pharmacological methods for the treatment of opioid-induced sexual dysfunction. However, as these drugs may be only one of the etiological factors, the use of standard non-pharmacological methods in these disorders should be considered. These include, among others, psychotherapy or psychosexual therapy [75,76], physiotherapy [77], physical therapy with the use of instruments [78] and the use of lubricants in women [75], and the use of erection aids in men [79].

## 6. Conclusions

Sexual disorders are common among cancer patients undergoing opioid therapy, yet they are frequently overlooked and untreated. The etiology of these disorders is multifactorial and, therefore, difficult to interpret, as the disease itself and its treatment could be confounding variables not controlled in the presented studies. However, despite the scarce evidence, opioids may be one of the etiological factors not widely known so far. The pathophysiology of this phenomenon is not yet clear and should be a subject for future research. It may include disorders of both the endocrine and nervous systems, resulting in, among other things, ED and deteriorating sexual desire, sexual arousal, orgasm, and general satisfaction with sex life. The treatment options for opioid-induced sexual dysfunction include testosterone replacement therapy, phosphodiesterase type 5 inhibitors, bupropion, opioid antagonists, and plant-derived means, such as damask rose oil and ginseng. However, most of them were studied in males, and further research on treatment for women is needed. Although opioids are believed to be an important cause, the etiology is usually multifactorial, so management should be multifaceted and individualized by targeting pathophysiological factors. One of the treatment options may also be a choice of an opioid that is less likely to cause sexual dysfunction, yet further research is necessary.

## Figures and Tables

**Figure 1 cancers-14-04046-f001:**
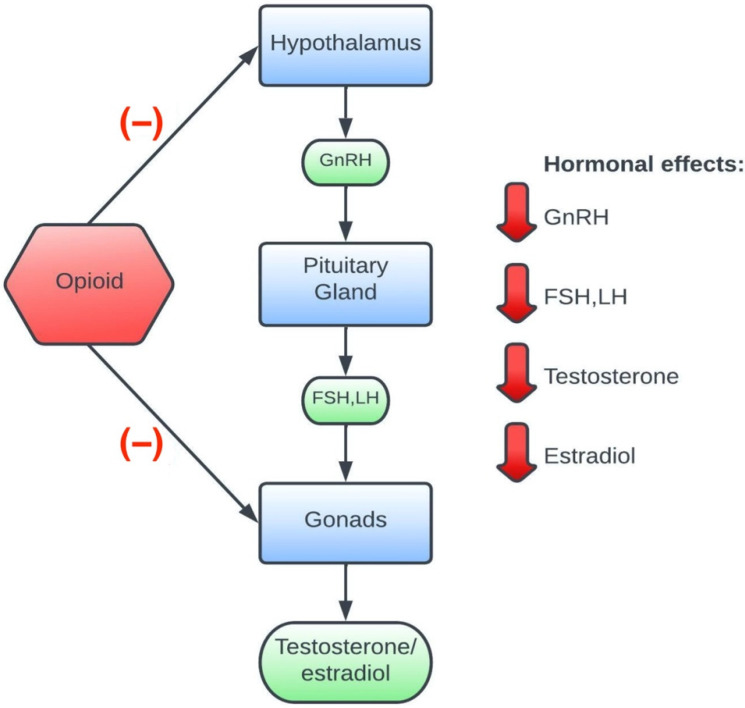
The impact of opioids on the hypothalamus−pituitary−gonadal axis.

**Table 1 cancers-14-04046-t001:** ICD-11 classification of sexual disorders possibly associated with opioids [7].

17 Conditions Related to Sexual Health
Sexual dysfunctions
HA00 Hypoactive sexual desire dysfunction
HA01 Sexual arousal dysfunctions
HA02 Orgasmic dysfunctions
HA03 Ejaculatory dysfunctions
HA0Y Other specified sexual dysfunctions
HA0Z Sexual dysfunctions, unspecified
Sexual pain disorders
HA20 Sexual pain-penetration disorder
HA2Y Other specified sexual pain disorders
HA2Z Sexual pain disorders, unspecified
HA40 Aetiological considerations in sexual dysfunctions and sexual pain disorders
HA40.2 Associated with use of psychoactive substance or medication

**Table 2 cancers-14-04046-t002:** Areas assessed by the diagnostic tools in women (Female Sexual Function Index) and men (International Index of Erectile Function) [39,43].

Female Sexual Function Index (FSFI)	International Index of Erectile Function (IIEF)
Desire Arousal Lubrication Orgasm Sexual satisfaction Pain	Erectile function Orgasmic function Sexual desire Intercourse satisfaction Overall satisfaction

**Table 3 cancers-14-04046-t003:** Pharmacological treatment options for opioid-induced sexual dysfunction.

Pharmacological Treatment Options
Testosterone Replacement Therapy Bupropion
Trazodone
Opioid antagonist (naltrexone, nalmefene)
Phosphodiesterase type 5 inhibitors
Plant-derived medicines (damask rose oil, ginseng)

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
