# Peer review of "Opioid-Induced Sexual Dysfunction in Cancer Patients"

_cancers, 2022, doi:10.3390/cancers14164046_

Round 1

Reviewer 1 Report

The present manuscript reports how opioids could play a role in the sexual dysfunction of cancer patients. Literature is scarce when it comes to sexual dysfunction in patients with cancer and the authors did a great job to cover all aspects that might be involved in this process and how it could be treated. 

Specific comments:

Only one concern that needs to be raised  in this manuscript, regarding the conclusion that states: "Despite the scarce evidence, opioids seem to be the primary etiological factor of these disorders, affecting up to half of opioid users."

Although the manuscript and the literature bring evidence that opioids may play a direct role in contributing to the development of sexual dysfunction, to state that this is the primary etiological factor is an overstatement, as sexual dysfunction can be also triggered by chemotherapy, radiotherapy, surgical procedures, psychological issues and more. Actually, studying sexual dysfunction in cancer patients undergoing opioid therapy would not be very easy to interpret as all other treatments and the disease itself could be a confounding variable. Therefore, the studies involving opioids in non-cancer patients seem to be very important to shed light on how opioids directly affect the psychological-neural-vascular-hormonal physiology of sexual function, extrapolating the interpretation to cancer patients. But the role of other cancer therapies (and the disease itself) in inducing sexual dysfunction cannot be neglected. 

Author Response

Thank you for your review and valuable comment. We strictly followed your remark and changed the conclusion accordingly:

The etiology of these disorders is multifactorial and, therefore, difficult to interpret, as the disease itself and its treatment could be confounding variables not controlled in the presented studies. However, despite the scarce evidence, opioids may be one of the etiological factors, not widely known so far.”

Reviewer 2 Report

This publication highlights an important and fairly novel topic in focusing on the role of opioids in sexual dysfunction patients with cancer. However, the paper also seems to focus on opioids and ignore the many other factors thought to contribute to sexual dysfunction in this population. This is a problem throughout the manuscript, in discussion pathophysiology in the introduction, in discussing papers that do not evaluate for confounding factors associated with sexual dysfunction and/or cancer, and in the broad-sweeping conclusions. While I think there is value in this paper, these issues need to be addressed.

-          Section 2 should highlight cancer-specific pathophysiology of sexual dysfunction, beyond mentioning “multifactorial”. Mentioning that illness, mental health, body image, chemotherapy, radiation, and surgery can affect things like hormones, drive, orgasm, pain, etc are critical.

-          The section on orgasm primarily defines orgasm and briefly mentions orgasmic disorder. No connection is made to opioids, or to frequency of problems in cancer. This should be edited to address what is known about these issues, or the section is irrelevant to the paper.

-          The diagnostic tools section is fairly generic. Are there tools that are better validated in cancer populations or populations of patients on opioids? Should there be additional questions asked about opioid use?

-          It seems likely that there are confounding factors when comparing patients taking and not taking opioids, such as pain, mental health, physical health, and quality of life. These factors could certainly affect sexual dysfunction. Further, addiction is very different from appropriate prescription opioid use. Have any of the studies mentioned in section 4.2 evaluated/attempted to adjust or control for this? If not, this should be mentioned as a potential confounding/limiting factor for these studies.

-          Has dilaudid been specifically evaluated? Since this is a commonly used opioid, it may be worth mentioning.

-          Are there studies evaluating sustained release vs immediate release opioids and sexual dysfunction?

-          Treatment of opioid-induced sexual dysfunction focuses on males. Are there studies of treatment for females? If not, this should still be mentioned as a gap in the literature

-          This statement seems baseless, ignoring many other believed etiologies of sexual dysfunction in cancer (mental health, physical health, fatigue, endocrine dysfunction, radiation, chemotherapy, pelvic surgeries): “Despite the scarce evidence, opioids seem to be the primary etiological factor of these disorders, affecting up to half of opioid users.” This is particularly true given most studies are not actually on patients with cancer, and none of the mentioned studies seemed to control for any other factors.

-          The conclusions list only treatments for males, ignoring females.

-          Are there papers evaluating rates of sexual dysfunction in cancer patients on and off of opioids? This would help to demonstrate if the effect is additive. If not, this should be mentioned as an important gap in literature.  

Author Response

Thank you very much for your detailed and fair review. In general, we accepted all your valuable remarks and followed them appropriately, as follows:

-          Section 2 should highlight cancer-specific pathophysiology of sexual dysfunction, beyond mentioning “multifactorial”. Mentioning that illness, mental health, body image, chemotherapy, radiation, and surgery can affect things like hormones, drive, orgasm, pain, etc are critical.

       This section has been expanded following your remark.

-       The section on orgasm primarily defines orgasm and briefly mentions orgasmic disorder. No connection is made to opioids, or to frequency of problems in cancer. This should be edited to address what is known about these issues, or the section is irrelevant to the paper.

      The data on the prevalence of orgasm disorders in cancer patients and the mechanisms of influence of opioids on orgasm is very limited. However, as it is a very important sexual function and the prevalence of opioid-induced dysfunctions of orgasm is mentioned in cited articles in a further part of the article we find it important to keep this part in our article.

    This section has been changed – the information on limited data has been added.

-         The diagnostic tools section is fairly generic. Are there tools that are better validated in cancer populations or populations of patients on opioids? Should there be additional questions asked about opioid use?

      The IIEF and FSFI are the most popular questionnaires, and they are recommended for the general population and cancer patients as well. Additionally, some tools were developed for patients after treatment for specific cancer - University of California-Los Angeles Prostate Cancer Index (UCLA PCI), Sexual Function-Vaginal Changes Questionnaire for the assessment after gynecological cancer, and FSFI adaption for breast cancer patients (FSFI-BC). There are no questionnaires specific to patients with sexual dysfunctions caused by opioids.

The section has been revised. We added a comment “None of the available tools refer to opioid use, nor have they been validated in the opioid users' population.”

-         It seems likely that there are confounding factors when comparing patients taking and not taking opioids, such as pain, mental health, physical health, and quality of life. These factors could certainly affect sexual dysfunction. Further, addiction is very different from appropriate prescription opioid use. Have any of the studies mentioned in section 4.2 evaluated/attempted to adjust or control for this? If not, this should be mentioned as a potential confounding/limiting factor for these studies.

       None of the studies compared confounding factors between opioid and non-opioid groups. The information about limiting factors has been added at the end of chapter 4.2.

-         Has dilaudid been specifically evaluated? Since this is a commonly used opioid, it may be worth mentioning.

      No publications are evaluating Dilaudid (hydromorphone) on sexual dysfunctions.

-          Are there studies evaluating sustained release vs immediate release opioids and sexual dysfunction?

       There is one study by Rubinstein et al. on long-acting (sustained release) vs. short-acting (immediate release) opioids, and it has been added in the  section 4.2.

-         Treatment of opioid-induced sexual dysfunction focuses on males. Are there studies of treatment for females? If not, this should still be mentioned as a gap in the literature

      There are only the cited studies (Farnia et al. 2017, Farnia et al. 2019) on damask oil and ginseng. The information on the gap in the literature and limitations of the presented studies has been added.

-         This statement seems baseless, ignoring many other believed etiologies of sexual dysfunction in cancer (mental health, physical health, fatigue, endocrine dysfunction, radiation, chemotherapy, pelvic surgeries): “Despite the scarce evidence, opioids seem to be the primary etiological factor of these disorders, affecting up to half of opioid users.” This is particularly true given most studies are not actually on patients with cancer, and none of the mentioned studies seemed to control for any other factors.

The mentioned statement has been changed to: “The etiology of these disorders is multifactorial and, therefore, difficult to interpret, as the disease itself and its treatment could be confounding variables not controlled in the presented studies. However, despite the scarce evidence, opioids may be one of the etiological factors, not widely known so far.”

-          The conclusions list only treatments for males, ignoring females.

       The information on the gap in the literature has been added.

-         Are there papers evaluating rates of sexual dysfunction in cancer patients on and off of opioids? This would help to demonstrate if the effect is additive. If not, this should be mentioned as an important gap in literature.

There is only one paper – the cited case-control study by Rajagopal et al.

Once again, we are grateful for your detailed and invaluable remarks. We hope our responses will meet your expectations.

Reviewer 3 Report

Critique

·         This statement on page2 “oxytocin reduces sexual arousal and the occurrence of refraction after orgasm” requires a little more clarification.  I do not have a good sense of what the author means

·         There needs to be a clarification of “sexual reaction (line 74 on page 2) is it “arousal” or “response”?

·         Does oxytocin increase or decrease the time to erection? (lines 74 page 6)

·         Opioids produce hypogonadotropic hypogonadism.

·         Heroin is diacetylmorphine which becomes morphine in circulation.  The sexual dysfunction with heroin is likely related to morphine.

·         One wonders about peripheral restricted opioid receptor antagonists as a treatment for opioid-induced sexual dysfunction.

·         In regard to treatment, the combination of a selective serotonin reuptake inhibitor plus an opioid is likely taken compound the problem with sexual dysfunction which is predominantly erectile dysfunction in males.  The choice of an antidepressant should be either mirtazapine or bupropion.

·         Interestingly tramadol has been used in multiple studies for premature ejaculation.

·         Paradoxically in 1 study (Ramdurg,2012) the opioid antagonist naltrexone produced greater erectile dysfunction than buprenorphine.

·         Other potential treatments for sexual dysfunction on opioids include rosa Damascena, trazodone, and ginseng ( Ramli, 2021)

·         iPDES also may improve sexual dysfunction in patients on opioids (Ajo 2017)

·         There is the question about the duration of treatment and in particular whether long-acting opioids are more likely to cause her hypogonadism and sexual dysfunction than short-acting opioids.  Perhaps a dosing strategy of as-needed short-acting opioids would maintain sexual function

·         Testosterone replacement may improve pain but not sexual function ( AminiLari 2019)

Author Response

Thank you for your invaluable comments. We followed them strictly, and beneath you will find our responses.

  • This statement on page2 “oxytocin reduces sexual arousal and the occurrence of refraction after orgasm” requires a little more clarification. I do not have a good sense of what the author means

       It has been changed to “Oxytocin at high doses probably reduces sexual arousal and causes a refraction period (feeling of sexual satiety and inhibited sexual behavior) after orgasm in males, whereas, at lower doses, it probably stimulates sexual behavior.”

  • There needs to be a clarification of “sexual reaction (line 74 on page 2) is it “arousal” or “response”?

       Arousal is the proper term here. It has been changed. (line 85 now)

  • Does oxytocin increase or decrease the time to erection? (lines 74 page 6)

      The cited study suggests that it probably plays a role in enabling the occurrence of erection, not referring to the impact on the time to the erection.

  • Opioids produce hypogonadotropic hypogonadism.

        It is one of the mechanisms of causing sexual dysfunctions by opioids mentioned in the article. Therefore, the term “hypogonadotropic hypogonadism” has been added in section 2.1.

  • Heroin is diacetylmorphine which becomes morphine in circulation. Therefore, the sexual dysfunction with heroin is likely related to morphine.

       This information has been added in section 4.2.2.

  • One wonders about peripheral restricted opioid receptor antagonists as a treatment for opioid-induced sexual dysfunction.

       There are no studies on peripheral restricted opioid receptor antagonists as a treatment for opioid-induced sexual dysfunction as a treatment option. Therefore, this information has been added in section 5.3.

  • In regard to treatment, the combination of a selective serotonin reuptake inhibitor plus an opioid is likely taken compound the problem with sexual dysfunction which is predominantly erectile dysfunction in males. The choice of an antidepressant should be either mirtazapine or bupropion.

       There are no studies on mirtazapine as a treatment for opioid-induced sexual dysfunction. The study of bupropion has already been included in the article.

The combination of a selective serotonin reuptake inhibitor and opioid is not recommended and should be avoided because of the risk of serotonin syndrome.

  • Interestingly tramadol has been used in multiple studies for premature ejaculation.

      This information has been added in section 4.2.1.

  • Paradoxically in 1 study (Ramdurg 2012) the opioid antagonist naltrexone produced greater erectile dysfunction than buprenorphine.

      In this study, erection difficulty was reported more often by patients in the naltrexone group vs. buprenorphine group (66,7% vs. 43,3%) when they were asked about experiencing erection difficulty “ever.” However, when asked about last month, both groups reported experiencing erection difficulty with similar frequency. Additionally, sexual functions were similar between the two groups when measured with BMSFI.

      This study has been added to the article.

  • Other potential treatments for sexual dysfunction on opioids include rosa Damascena, trazodone, and ginseng ( Ramli, 2021)

      The article has already included the studies on rosa damascena and ginseng. In addition, the study about trazodone has been added.

  • iPDES also may improve sexual dysfunction in patients on opioids (Ajo 2017)

       This study has already been included in the article. The information about PDE5i has been added to table 3.

  • There is the question about the duration of treatment and in particular whether long-acting opioids are more likely to cause her hypogonadism and sexual dysfunction than short-acting opioids. Perhaps a dosing strategy of as-needed short-acting opioids would maintain sexual function

      There is one study by Rubinstein et al. on long-acting vs. short-acting opioids, and it has been added in section 4.2.

       Dosing as-needed is not recommended for chronic opioid use by cancer patients (Management of cancer pain in adult patients: ESMO Clinical Practice Guidelines 2018; WHO Guidelines for the pharmacological and radiotherapeutic management of cancer pain in adults and adolescents 2018)

  • Testosterone replacement may improve pain but not sexual function ( AminiLari 2019)

       The results of this study have been included in the article.

Round 2

Reviewer 3 Report

The revised manuscript is better. No additional comments